# Review on Type of Sensors and Detection Method of Anti-Collision System of Unmanned Aerial Vehicle

**DOI:** 10.3390/s23156810

**Published:** 2023-07-30

**Authors:** Navaneetha Krishna Chandran, Mohammed Thariq Hameed Sultan, Andrzej Łukaszewicz, Farah Syazwani Shahar, Andriy Holovatyy, Wojciech Giernacki

**Affiliations:** 1Laboratory of Biocomposite Technology, Institute of Tropical Forestry and Forest Products (INTROP), University Putra Malaysia, Serdang 43400, Selangor Darul Ehsan, Malaysia; navaneethakrishna93@gmail.com; 2Department of Aerospace Engineering, Faculty of Engineering, University Putra Malaysia, Serdang 43400, Selangor Darul Ehsan, Malaysia; farahsyazwani@upm.edu.my; 3Aerospace Malaysia Innovation Centre (944751-A), Prime Minister’s Department, MIGHT Partnership Hub, Jalan Impact, Cyberjaya 63000, Selangor Darul Ehsan, Malaysia; 4Institute of Mechanical Engineering, Faculty of Mechanical Engineering, Bialystok University of Technology, 15-351 Bialystok, Poland; 5Department of Computer-Aided Design Systems, Lviv Polytechnic National University, 79013 Lviv, Ukraine; andrii.i.holovatyi@lpnu.ua; 6Institute of Robotics and Machine Intelligence, Faculty of Control, Robotics and Electrical Engineering, Poznan University of Technology, 60-965 Poznan, Poland; wojciech.giernacki@put.poznan.pl

**Keywords:** anti-collision methods, detection system, sensors, unmanned aerial vehicle

## Abstract

Unmanned aerial vehicle (UAV) usage is increasing drastically worldwide as UAVs are used in various industries for many applications, such as inspection, logistics, agriculture, and many more. This is because performing a task using UAV makes the job more efficient and reduces the workload needed. However, for a UAV to be operated manually or autonomously, the UAV must be equipped with proper safety features. An anti-collision system is one of the most crucial and fundamental safety features that UAVs must be equipped with. The anti-collision system allows the UAV to maintain a safe distance from any obstacles. The anti-collision technologies are of crucial relevance to assure the survival and safety of UAVs. Anti-collision of UAVs can be varied in the aspect of sensor usage and the system’s working principle. This article provides a comprehensive overview of anti-collision technologies for UAVs. It also presents drone safety laws and regulations that prevent a collision at the policy level. The process of anti-collision technologies is studied from three aspects: Obstacle detection, collision prediction, and collision avoidance. A detailed overview and comparison of the methods of each element and an analysis of their advantages and disadvantages have been provided. In addition, the future trends of UAV anti-collision technologies from the viewpoint of fast obstacle detection and wireless networking are presented.

## 1. Introduction

General Visual Inspection (GVI) is a typical approach for quality control, data collection, and analysis. It involves using basic human senses such as vision, hearing, touch, smell, and non-specialized inspection equipment. Unmanned aerial systems (UAS), also known as UAVs, are being developed for automated visual inspection and monitoring in various industrial applications [1]. These systems consist of UAVs outfitted with the appropriate payload and sensors for the job at hand [2].

Sensor and measurement reliance is crucial for UAV operations and functionality, as they serve as indispensable resources to ensure the safety and security of UAVs. Since UAVs operate autonomously without a pilot’s input, a series of sensors and systems are required for the UAVs to position themselves. Usually, UAVs use a global positioning system (GPS) to position themselves. However, GPS input will not always be accurate, especially when the UAV has to be equipped with sensors such as rangefinders, which are very useful when the UAV flies at low altitudes. The investigation of the quadcopter control problem came to a standstill until relatively recently, since the control of four separate motor-based propulsion systems was nearly impossible without modern electronic equipment. These technologies have only become increasingly sophisticated, versatile, quick, and affordable in the past several decades. 

Due to the intricacy of the issue, controlling a quadcopter is a topic that is both intriguing and important. The fact that the system has just four inputs (the angular velocity of the propellers) despite having six degrees of freedom (three rotational axes and three transnational axes) gives the system the quality of being under-actuated [3]. Even though some of them have more than six inputs, they all have the same number of axes to manipulate, meaning they are all under-actuated. This is because all those inputs can only directly control the three rotation axes, not the translation axis [4].

Additionally, the dynamics on which this form of UAV operates give freedom in movement and robustness towards propulsion problems. This sort of UAV is ideal for reconnaissance missions. As an illustration, control algorithms may be programmed so that a UAV can keep its stability even if fifty percent of the propellers that control one axis of rotation stop working correctly. On the other hand, since it is an airborne vehicle, the frictions of the chassis are almost non-existent, and the control algorithm is responsible for handling the damping. 

A UAV’s level of autonomy is defined by its ability to perform a set of activities without direct human intervention [5]. Different kinds of onboard sensors allow unmanned vehicles to make autonomous decisions in real time [6,7,8]. Demand for unmanned vehicles is rising fast because of the minimal danger to human life, enhanced durability for more extended missions, and accessibility in challenging terrains. Still, one of the most difficult problems to address is planning their course in unpredictable situations [9,10,11]. The necessity for an onboard system to prevent accidents with objects and other vehicles is apparent, given their autonomy and the distances they may travel from base stations or their operators [12,13].

Whether a vehicle is autonomous or not, it must include a collision avoidance system. Several potential causes of collisions include operator/driver error, machinery failure, and adverse environmental factors. According to statistics provided by planecrashinfo.com, over 58% of fatal aviation crashes occurred due to human mistakes between January 1960 and December 2015 [14]. To reduce the need for human input, the autopilot may be upgraded with features like object recognition, collision avoidance, and route planning. Methods of intelligent autonomous collision avoidance have the potential to contribute to making aircrafts even safer and saving lives.

The exponential growth in UAVs using in public spaces has made a necessity for sophisticated and highly dependable collision avoidance systems evident and incontestable from the public safety perspective. UAVs can access risky or inaccessible locations without risking human lives. Therefore UAVs should be built to operate independently and avoid crashing into anything while in flight [15]. Precision agriculture is an application of UAVs that has been increasing rapidly worldwide. Precision agriculture is expanding quickly in commercial goods and research and development applications. In order to correctly account for the geographical and temporal fluctuations of crop and soil components, this revolutionary trend is redefining the crop management system and placing a higher focus on data collecting and analysis, whether in real-time or offline.

Figure 1 shows the basic architecture of an anti-collision system that will be implemented in a vehicle. Anti-collision systems consist of two major parts: the input and output [15]. These parts can also be recognized as perspective and action. Any system designed to prevent accidents from happening must begin with perception, or more specifically, obstacle detection [16]. At this stage, sensors gather information about the surrounding area and locate any hazards. However, the active part comes after the perspective, where once the threat has been detected, the situation will be analyzed by the computation of the control system of the UAVs. As a result, the actuators will implement proper countermeasures to avoid the hazard [17].

Sensors come in a wide variety, but they may be broken down into two broad categories: active and passive. The backscatter is measured by an active sensor with its own source that sends out a beam of light or a wave. On the other hand, passive sensors can only estimate the energy emitted by an item, such as sunlight reflected off the object. Anti-collision systems use a total of four different approaches in detecting the hazards, which are geometric (using the UAV’s and obstacles’ positions and velocities to reformat nodes, typically via trajectory simulation), force-field (manipulating attractive and repulsive forces to avoid collisions), optimized (using the known parameters of obstacles to find the most efficient route), and sense-and-avoid (making avoidance decisions at runtime based on sensing the environment) [18,19].

The complexity of collision avoidance systems may vary from as simple as alerting the vehicle’s pilot to be involved to wholly or partly taking control of the system on its own to prevent the accident [20]. For an unmanned vehicle to travel without direct human intervention, it must be equipped with several specialized systems that identify obstacles, prevent collisions, plan routes, determine their exact location, and implement the necessary controls [21]. Multiple UAVs provide substantial benefits over single UAVs. They are in high demand for a wide range of applications, including military and commercial usage, search and rescue, traffic monitoring, threat detection (particularly near borders), and atmospheric research [22,23,24]. UAVs may struggle to complete missions in a demanding dynamic environment due to cargo restrictions, power constraints, poor vision due to weather, and difficulties in remote monitoring. To ensure unmanned vehicles’ success and safe navigation, the robotics community is working tirelessly to overcome these difficulties and deliver the technical level fit for challenging settings [25,26,27,28]. 

One of the most challenging problems for autonomous vehicles is detecting and avoiding collisions with objects, which becomes much more critical in dynamic situations with several UAVs and moving obstacles [29]. Sensing is the initial process in which the system takes data from its immediate environment. When an impediment enters the system’s field of view, the detection stage performs a risk assessment. To prevent a possible collision, the collision avoidance module calculates how much of a detour has to be made from the original route. Once the system has completed its calculations, it will execute the appropriate move to escape the danger safely.

## 2. Obstacle Detection Sensors

The drone needs a “perspective model” of its environment to avoid crashing into obstacles [30,31]. To do this, the UAV must have a perception unit consisting of one or more sensors [32]. Sensors, like imaging sensors of varying resolutions, are crucial components of remote sensing systems. Sensors may be used in a wide variety of contexts. LiDAR, visible cameras, thermal or infrared cameras, and solid-state or mechanical devices are all examples of sensors that may be used for monitoring [27,33]. The sensors that have been used for the anti-collision system are majorly categorized into two, which are active sensors and passive sensors. In Figure 2, the categorization of the anti-collision system sensors is shown.

### 2.1. Active Sensors

Sensing using active sensors involves emitting radiation and then detecting the reflected radiation. All the necessary components, including the source and the detector, are built within an active sensor. A sensor works by having a transmitter send out some signal (light, electricity, sound) that then gets reflected off of whatever it is being used to detect [34,35]. Most of these sensors operate in the spectrum’s microwave range, allowing them to penetrate the atmosphere under most circumstances. The metrics of interest of the obstacles, such as distance and angles, may be adequately returned by such sensors since they have a short reaction time, need less processing power, can scan more significant regions quickly, and are less impacted by weather and lighting conditions. In [36], the authors use MMW radar. In their setup, things are detected and followed by watching radar echoes and figuring out how far away they are from the vehicle. Different distances and weather conditions are also used to conclude the performance. Despite the allure, radar-based solutions are either too costly or too heavy to be practical on more miniature robots, such as battery-powered UAVs [37,38].

#### 2.1.1. Radar

A radar sensor transmits a radio wave that will be reflected back to the sensor after hitting an object. The distance between the object and the radar is determined by timing how long it takes the signal to return. Despite their high cost, airborne radar systems are often used for their precision to provide data. Both continuous-wave and pulsed-wave radars exist, with the former emitting a steady stream of linearly modulated (or frequency-modulated) signals and the latter emitting intense but brief bursts of signals; however, both types have blind spots [39]. As a bonus, radars could also track the objects’ speeds and other motion data. For instance, the radar may determine an object’s velocity by measuring how much the frequency of its echo or bounced-off signal changes as it approaches the radar [40].

Using a compact radar, the authors of [40] could get range data in real time, regardless of the weather. The system incorporates a compact radar sensor and an OCAS (obstacle collision avoidance system) computer. OCAS utilizes radar data such as obstacle velocity, azimuth angles, and range to determine avoidance criteria and provide orders to the flight controller to execute the appropriate maneuver to prevent collisions. The findings indicated that with the set safety margins, the likelihood of successfully avoiding a crash is more than 85%, even if there is an inaccuracy in the radar data.

The benefits of integrating radar sensors into UAVs for obstacle identification and for detecting and calculating additional aspects of the observed obstruction, such as the velocity of the obstacle and the angular information utilizing multichannel radars, are thoroughly explored by the authors in [41]. Experiments reveal that with forward-looking radars, with the radar’s simultaneous multi-target range capabilities, it is possible to identify targets across an extensive angular range of 60 degrees in azimuth. For their suggested autonomous collision avoidance system, the authors of [41] used Ultra-Wideband (UWB) collocated MIMO radar. Radar cognition’s capacity to modify the waveform of ultra-wideband multiple-input multiple-output radar transmissions for better detection and, by extension, to steer the UAV by giving an estimate of the collision locations is a significant advantage.

#### 2.1.2. LiDAR

One may compare the operation of a light detection and ranging (LiDAR) sensor to that of a radar. One half of a LiDAR sensor fires laser pulses at the surface(s), while the other half scans their reflection and calculates distance based on how long each pulse takes to return. Rapid and precise data collection is achieved using LiDAR. LiDAR sensors have shrunk in size and shed weight over the years, making it possible to put them on mini and small UAVs [42,43]. LiDAR-based systems are more cost-effective than radar systems, particularly those using 1D and 2D LiDAR sensors.

The designed system was successfully field tested by the authors of [44] using a variety of laser scanners installed on a vehicle, which are laser radars ranging in three dimensions. Regarding 3D mapping and 3D obstacle detection, 3D LiDARs are as standard as it gets in the sensor world [45,46]. Since LiDAR is constantly being moved and ranged, the gathered data is prone to motion distortion, which makes using these devices challenging. To get around this, as proposed by the authors of [45], additional sensors may be used with LiDAR. Only 3D LiDARs allow for precise assessment of an object’s posture.

#### 2.1.3. Ultrasonic

To determine an item’s distance, ultrasonic sensors transmit sound waves and then analyze the echoes they receive [47]. The sound waves produced are outside the range humans can hear (25 to 50 kilohertz) [48]. Compared to other types of range sensors, ultrasonic sensors are both more affordable and widely accessible. The object’s transparency does not affect ultrasonic sensors, unlike LiDARs. Unlike ultrasonic sensors, which are color-blind, LiDARs have trouble identifying transparent materials like glass. However, the sonic sensor will not provide accurate readings if the item reflects the sound wave in the opposite direction than the receiver or if the substance has the properties of absorbing sound.

Like radars and LiDARs, this method relies on emitting a wave, waiting for the reflected wave to return, and then calculating the distance based on the time difference between the two. Compared to other types of range sensors, ultrasonic sensors are both more accessible and more affordable. Since each sensor in Table 1 has its advantages and disadvantages compared to the others, it is clear that more than one sensor can be employed to provide complete protection against the collision avoidance issue. Multiple sensors may be utilized to cover a greater area and eliminate blind spots, or different kinds of sensors can be fused to create a super sensor whose weaknesses cancel out those of its components.

According to Table 1, the LiDAR and ultrasonic sensors, which can be used in the UAV’s anti-collision system, are smaller than radar. This makes the ultrasonic and LiDAR the ideal method of obstacle sensing for small UAVs, as they are less in weight, reducing the UAV’s payload. In addition, the power consumption by ultrasonic and LiDAR is also low compared to radar. However, the accuracy and range of the radar are highest compared to ultrasonic and LiDAR, which makes the radar suitable for use in large UAVs that fly at high altitudes. On the other hand, the radar is not affected by weather conditions, but the LiDAR is affected, while ultrasonic is slightly affected by the weather condition. Last but not least, the cost of an ultrasonic sensor is the lowest compared to radar and LiDAR, which makes it more affordable.

### 2.2. Passive Sensors

The energy the seen items or landscape gives off is measured using passive sensors. Optical cameras, infrared (IR) cameras, and spectrometers are the most common types of passive sensors now used in sensing applications [49]. Wide varieties of cameras, each optimized for a specific wavelength, exist. The authors of [50] offer a system for acoustic signal tracking and real-time vehicle identification. The result is obtained by isolating the resilient spatial characteristics from the noisy input and then processing them using sequential state estimation. They provide empirical acoustic data to back up the suggested technique.

In contrast, thermal or infrared cameras operate in the infrared light range and have a larger wavelength than the visible light range. Therefore, the primary distinction between the two is that visual cameras use visible light to create a picture, while thermal cameras use infrared radiation. Ordinary cameras struggle when light levels are low, while IR cameras thrive [51]. It takes more computational resources since an additional algorithm is required to extract points of interest in addition to the algorithm already needed to calculate the range and other characteristics of the barriers [52]. Vision cameras are susceptible to environmental factors, including sunlight, fog, and rain, in addition to the field-of-view restrictions imposed by the sensor being employed [53,54].

#### 2.2.1. Optical

Taking pictures of the world around us is the foundation of visual sensors and cameras, which then utilize those pictures to extract information. There are three main types of optical cameras: monocular, stereo, and event-based [55,56,57]. Using cameras has several advantages, including their compact size, lightweight, low power consumption, adaptability, and simple mounting. Some drawbacks of employing such sensors include their sensitivity to lighting and background color changes and their need for clear weather. When any of these conditions are present, the recorded image’s quality plummets, significantly influencing the final product.

According to [58], a monocular camera may be used to identify obstacles in the path of a ground robot. Coarse obstacle identification in the bottom third of the picture is achieved by an enhanced Inverse Perspective Mapping (IPM) with a vertical plane model; however, this method is only suitable for slow-moving robots. Using stereo cameras is one method proposed by the authors of [59]. In stereo cameras, absolute depth is determined by combining internal and external camera characteristics, unlike in monocular cameras. The amount of processing power needed rises when stereo images are used. Because of the high processing cost and the need to accommodate highly complex systems with six degrees of freedom, like drones, the authors solve this problem by dividing the collected pictures into nine zones.

#### 2.2.2. Infrared

Sensors operating in the infrared spectrum, such as those used in infrared (IR) cameras, are deployed when ambient light is scarce. They may also be used with visual cameras to compensate for the latter’s lackluster performance, particularly at night. Data from a thermal camera may be analyzed by automatically determining the image’s orientation by extracting fake control points due to the thermal camera’s output being hazy and distorted with lesser resolution than that of an RGB camera [60].

## 3. Obstacle Detection Method

Both reactive and deliberative planning frameworks may be used for collision avoidance. During management by reaction, the UAV is equipped with onboard sensors to collect data about its immediate environment and behave accordingly. It facilitates instantaneous responses to changing environmental conditions. An alternative navigational strategy may be necessary if reactive control leads to a local minimum and becomes trapped there. The method of decision-making used by autonomous commercial cars will determine their level of safety and sanity. By dynamically connecting rear anti-collision elements, a driving decision network built on an actor-critic architecture has been developed to ensure safe driving. To interpret sensor data efficiently, this network considers the effects of different elements on collision prevention, such as rearward target detection, safety clearance, and vehicle roll stability. This has been accomplished by creating an improved reward function that considers these factors inside a multi-objective optimization framework. The network attempts to improve collision avoidance skills and guarantee the safety and stability of the vehicle by thoroughly examining these parameters. The force-field method, geometry, optimization-based methods, and sense-and-avoid techniques are the four main approaches to collision avoidance algorithms, as shown in Figure 3.

### 3.1. Force-Field Method

Using the idea of a repulsive or attractive force field, force-field techniques (also called potential field methods) may steer a UAV away from obstruction or draw it closer to a target [61,62]. Instead of using physical barriers, the authors of [63] propose using a potential field to surround a robot. In order to determine the shortest route between two places, the authors of [64] suggest using an artificial potential field. The points that create repulsive and attractive pressures for the robot are the obstacles and the targets, respectively.

The authors of [65] suggested a new artificial potential field technique to generate optimum collision-free paths in dynamic environments with numerous obstacles, where other UAVs are also treated as moving obstacles. This method is dubbed an improved curl-free vector field. Although simulations confirmed the method’s viability, more validation in 3D settings with static and dynamic factors is required [66]. Regarding UAV navigation in 3D space, the authors of [67] describe an artificial potential field technique that has been improved to produce safe and smooth paths. By factoring in the behavior of other UAVs and their interactions, the proposed optimized artificial potential field (APF) algorithm improves the performance of standard APF algorithms. During route planning, the algorithm considers other UAVs to be moving obstacles.

A vehicle collision avoidance algorithm is provided in [68], using synthetic potential fields. The algorithm considers the relative velocities of the cars and the surrounding traffic to decide whether to slow down or speed up to pass another vehicle. This decision is based on the size and the form of the potential fields of the barriers. Too big of a time step might lead to collisions or unstable behavior, so getting it exactly right is essential. A 1D virtual force field approach is proposed for moving obstacle detection [69]. They argue that the inability to account for the barriers’ mobility causes the efficiency loss seen with conventional obstacle force field approaches.

### 3.2. Sense and Avoid Method

In order to control the flight path of each UAV in a swarm without information about the plans of other drones, with fast response time, sense-and-avoid techniques focus on reducing the computational power required by simplifying the collision avoidance process to individual detection and avoidance of obstacles. Methods based on “Sense and Avoid” The speed with which collision avoidance can respond makes it a good tool for complex contexts. A robot or agent is outfitted with several sensing technologies, including LiDAR, sonar, and radar. Although it cannot distinguish between different objects, radar can quickly respond to anything that enters its field of view [69,70,71].

In [72], the authors suggest a technique for categorizing objects as static or dynamic using 2D LiDAR data. Additionally, the program can provide rough estimates of the speeds of the moving obstructions. In [73], the authors use a computer vision method to implement an animal detection and collision-avoidance system. The team has trained its system with over 2200 photos and tested it with footage of animals in traffic. In [74], the authors implement a preset neural network module in MATLAB to operate with five ultrasonic (US) sensors to triangulate and determine objects’ exact location and form. They use three distinct shapes in their evaluations. To accomplish object recognition and avoidance, the inventors of [75] fused a US sensor with a binocular stereo-vision camera. Using stereo vision as the primary method, a new route is constructed via an algorithm based on the Rapidly Explored Random Tree (RRT) scheme.

### 3.3. Geometric Method

To ensure that the predetermined minimum distances between agents, such as UAVs, are not violated, geometric techniques depend on studying geometric features. The UAVs’ separation distances and travel speeds have been used to calculate the time remaining until a collision occurs. In [76], the authors provide an analytical method for resolving the planar instance of the issue of aircraft collision. We can find closed-form analytical solutions for the best possible sets of orders to end the dispute by analyzing the trajectories’ geometric properties.

In [77], conflict avoidance in a 3D environment is accomplished by using information such as the aircraft’s coordinates and velocities in conjunction with a mixed geometric and collision cone technique. However, the authors depend on numerical optimization techniques for the most common scenarios and only get analytical conclusions for specific circumstances. The paper [78] investigates UAV swarms that use geometry-based collision avoidance techniques. The suggested method integrates line-of-sight vectors with relative velocity vectors to consider a formation’s dynamic limitations. Each UAV may assess if the formation can be maintained while avoiding collisions by computing a collision envelope and using that information to determine the potential directions for avoiding collisions.

In [79], the authors combined geometric avoidance and the selection of start time from critical avoidance to provide a novel approach to collision avoidance based on kinematics, the risk of collisions, and navigational constraints. Instead of trying to avoid all of the barriers simultaneously, FGA may prioritize which obstacles must be avoided first, depending on how much time must pass before they can be safely passed. The authors of [80] developed a way to safely pilot UAVs from the beginning of a mission to its completion while ensuring that the vehicles stay on their intended course and avoid potential hazards. The authors offer a solution that individually tackles the system’s collision avoidance control and trajectory control and then merges them via a planned movement strategy.

### 3.4. Optimization Method

Methods based on optimization need geospatial data for the formulation of the avoidance trajectory. Probabilistic search algorithms aim to offer the most productive locations to conduct a search, given the level of uncertainty associated with that information. Different optimization techniques, such as those inspired by ants, genetic algorithms, gradient descent-based approaches, particle swarm optimization, greedy methods, and local approximations, have been developed to handle the enormous computing demands of these algorithms.

For instance, to successfully calculate optimum collision-free search pathways for UAVs under communication-related limitations, the authors of [81] use a minimum time search method with ant colony optimization. The authors of [82] provide a prediction technique for the next UAV coordinates based on the set of probable instructions the UAV will execute in the near future. After considering the destination coordinates and the UAV’s current location, the algorithm generates a cost function for the best trajectory. Using particle swarm optimization, a novel technique for autonomous vehicle route planning in the wild. This strategy uses the sensor data by giving various kinds of territory different weights, then using those weights to categorize the possible paths across the landscape.

### 3.5. Summary of Object Detection Method

Table 2 summarizes previous research studies on detection and anti-collision system. From Table 2, it can be concluded that the geometric detection and force field methods are suitable for long-range UAVs. However, the sense and avoid method is suitable for short-range UAVs. The compatibility of real-time detection in four detection methods allows the UAVs to analyze the surroundings and be more varied about the surrounding. The 3D compatibility in geometric, optimization, and sense-and-avoid methods allows the system to generate a 3D mapping around the surroundings, allowing the maneuvering to be more precise in the UAVs.

Other than these obstacle detection methods, which involve their implementation, many obstacle detection methods are being developed around the world. One obstacle detection method is neural network-based navigation. Human decisions about these types of motions may be observed in various situations, including those with randomly produced barriers and pertinent environmental data [84]. In comparison to human decision-making, the simulation results showed that the suggested method had a high estimation accuracy rate of almost 90%. In contrast to the adaptive project framework (APF) method, the neural network methodology demonstrated its usefulness by successfully navigating over obstacles without running into the local minimum problem, hence emphasizing the strength of neural network decision-making.

## 4. Conclusions

Analyzing this short review on the sensor type and detection method of anti-collision systems of UAVs, the selection of sensors and detection method mainly depends on the UAV type and the objective of the UAV mission. The table below presents the research gap and the stigmatization of the research review identified through the literature review.

In this context, the recommended method of detection in an anti-collision system in a UAV depends on the UAV’s specification and the UAV’s mission objective. Methods of obstacle detection using geometric are considered effective, where they are capable of 3 dimensions projection alternate route generation, and multiple UAV compatibility. However, they cannot communicate with ground control. The geometric object detection method basically uses input from GPS in order to position the UAV itself. This detection method is suitable in urban areas, where there will be strong GPS signals. However, strong GPS signals may not be found in rural areas, especially in plantation areas, where UAVs’ applications have rapidly increased in agricultural applications. When the GPS signal strength is low, the UAV cannot position itself accurately. Hence, optimization and sense and avoid methods will be more suitable in this case than geometric object detection methods. More specifically, optimization and sense-and-avoid detection methods are suitable for UAVs that fly at low altitudes; however, the geometric is ideal for high-altitude and long-range UAVs.

On the other hand, the force field detection method is more suitable in an environment consisting of multiple UAVs, where the UAV can sense the electromagnetic emission from other UAVs. However, although the force field method is the same as the geometric method, where it is suitable for long-range UAVs, it is not suitable for urban areas because there will be a lot of electromagnetic wave interference, eventually affecting the force field detection method. This literature review gives a better understanding of the anti-collision system within a UAV. It allows the optimization of anti-collision systems according to the UAV in which the anti-collision system will be implemented.

## Figures and Tables

**Figure 1 sensors-23-06810-f001:**
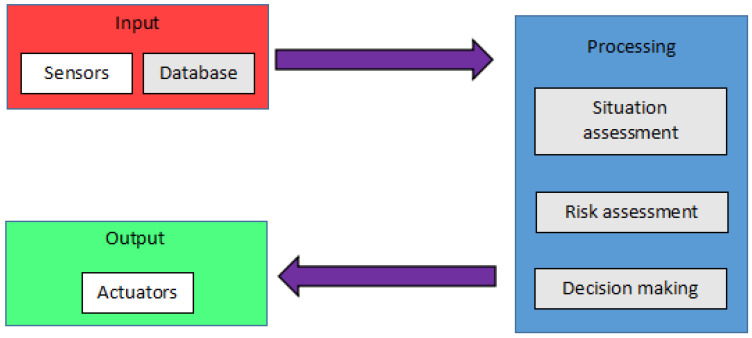
Anti-collision system general architecture.

**Figure 2 sensors-23-06810-f002:**
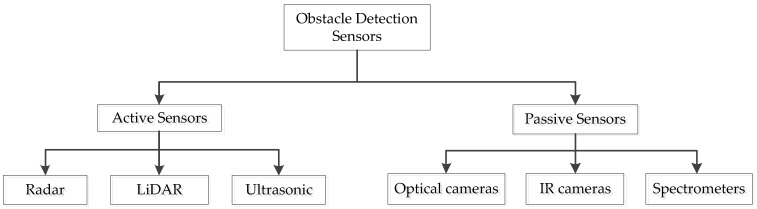
Categorization of anti-collision system sensors.

**Figure 3 sensors-23-06810-f003:**
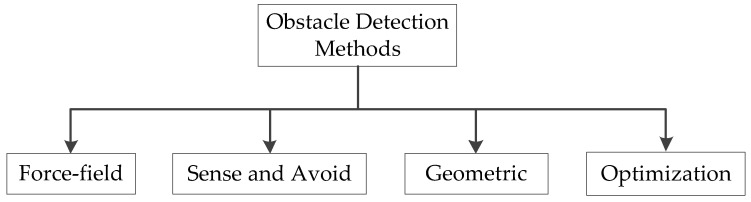
The main approaches to collision avoidance algorithms.

**Table 1 sensors-23-06810-t001:** Comparison between the active sensors of the anti-collision system.

Sensor	Sensor Size	Power Required	Accuracy	Range	Weather Condition	Light Sensitivity	Cost
Radar	Large	High	High	Long	Not Affected	No	High
LiDar	Small	Low	Medium	Medium	Affected	No	Medium
Ultrasonic	Small	Low	Low	Short	Slightly Affected	No	Low

**Table 2 sensors-23-06810-t002:** Previous studies of detection and anti-collision system.

	Geometric			Sense and Avoid		Force Field		Optimization
	[78,79]	[80]	[83]	[72]	[74]	[69]	[65]	[82]
Multiple UAV Compatibility	**/**	**/**	**/**	**/**	**/**	**/**	O	**/**
3D Compatibility	**/**	**/**	**/**	**/**	**/**	O	O	**/**
Communication	O	**/**	**/**	**/**	**/**	O	O	**/**
Alternate Route Generation	**/**	**/**	**/**	**/**	O	**/**	**/**	**/**
Real-time Detection	**/**	**/**	**/**	**/**	**/**	**/**	**/**	**/**

**/**—Available. O—Not Available.

## Data Availability

Data sharing does not apply to this article as no new data were created or analyzed in this study.

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
