# Peer review of "Review on Type of Sensors and Detection Method of Anti-Collision System of Unmanned Aerial Vehicle"

_sensors, 2023, doi:10.3390/s23156810_

Round 1
Reviewer 1 Report
The results of the study should be discussed in much more detail. In addition, I think it would be beneficial to discuss and add to the study much more than the studies on this subject in the introduction. Sensor types need to be addressed in more detail together with the problems used in the literature.
no comment
Reviewer 2 Report
The article received for review is interesting and topical, but I believe it needs major revision as it addresses issues at the level of principles, not fundamental issues that ensure the safe operation of UAVs. The bibliographic references used by the authors are not among the essential ones that define the broad issues of UAV safety. For example, I recommend the authors to consult the following paper ``Sensors and Measurements for UAV Safety: An Overview`` by Eulalia Balestrieri, Pasquale Daponte, Luca De Vito, Francesco Picariello and Ioan Tudosa, Sensors 2021, 21, 8253. https://doi.org/10.3390/s21248253.
Reviewer 3 Report
This paper propose a review on type of sensors and detection method of anti-collision system of unmanned aerial vehicle. In general, this topic is interesting and some comments are given as follows:
1. Figure 1 shows the basic architecture of an anti-collision system. However, Introduction lacks an in-depth explanation of this structure, particularly regarding the part on processing.
2. The title of Section 3 is “Obstacle Detection Methods”, but its main content focuses on obstacle avoidance methods, which deviates from the main topic of this paper.
3. The content of Table 2 is confusing. To provide clarity, please provide a detailed explanation of the meaning of each element.
4. Deep learning and reinforcement learning are widely recognized as popular and effective methods used for data processing and obstacle avoidance. To enhance the comprehensiveness of this review, it is advisable to incorporate additional relevant content.
5. The term “unmanned aerial vehicles” can be replaced by its abbreviations “UAV” in this paper.
6. Please update the references and incorporate the most recent research on anti-collision system.
7. Some other collision avoidance methods like neural network are suggested to be added in the literature review part, e.g., A Neural Network-Based Navigation Approach for Autonomous Mobile Robot Systems, Applied Sciences 12 (15), 7796
Minor editing of English language is required.
Reviewer 4 Report
The review is interesting because it investigates the type of UAVs and sensors, but some minor revisions need to be made:
- Introduction is articulate but paragraphs should be included on the use of these UAV drones and sensors in various fields, such as precision agriculture (see article (Pallottino, F., Antonucci, F., Costa, C., Bisaglia, C., Figorilli, S., & Menesatti, P. (2019). Optoelectronic proximal sensing vehicle-mounted technologies in precision agriculture: A review. Computers and Electronics in Agriculture, 162, 859-873).
- The table in the conclusions should be moved, perhaps to the paragraphs above where it reviews the types of UAVs.
- Pictures of the sensors or UAVs considered should be added in the body of the text, because there is no picture, and you could also include a table where you put the bibliography reviewed by sensor and UAV type.
- Review the formatting of the bibliography because some things like bold and italics are missing.
Round 2
Reviewer 1 Report
It is acceptable now
Reviewer 2 Report
The authors of the article were responsive and addressed the comments submitted. In these condition the paper meets the requirements to be published.
Reviewer 3 Report
The authors have addressed all my comments.